# Early childhood risk factors for constipation and soiling at school age: an observational cohort study

Jon Heron,[1] Mariusz Grzeda,[1] David Tappin,[2] Alexander von Gontard,[3] Carol Joinson[1]

► Additional material is published online only. To view please visit the journal online (http://dx.doi.org/10.1136/bmjpo-2017-000230).

[1]Department of Population Health Sciences, Bristol Medical School, University of Bristol, Bristol, UK
[2]Department of Child Health, School of Medicine, Scottish Cot Death Trust, University of Glasgow, Glasgow, UK
[3]Department of Child and Adolescent Psychiatry, Saarland University Hospital, Homburg, Germany

**Correspondence to**
Dr Jon Heron; jon.heron@bristol.ac.uk

## ABSTRACT

**Objective** Constipation and soiling are common in childhood. This study examines the comorbidity between childhood constipation and soiling and early childhood risk factors for these problems.

**Design** The sample comprised 8435 participants from the Avon Longitudinal Study of Parents and Children with maternally reported measures of constipation (six time points between 4 and 10 years) and soiling (five time points between 4 and 9 years). We used latent class analysis to extract longitudinal patterns of constipation and soiling. We examined whether the latent classes are differentially associated with maternally reported risk factors in early childhood (stool consistency, breast feeding, socioeconomic background, gestation, birth weight, developmental level and age at initiation of toilet training) using multinomial logistic regression models.

**Results** We extracted four latent classes: 'normative' (74.5%: very low probability of constipation or soiling), 'constipation alone' (13.2%), 'soiling alone' (7.5%) and 'constipation with soiling' (4.8%). Hard stools at 2½ years were associated with increased odds of constipation alone. Developmental delay at 18 months was associated soiling alone and constipation with soiling, but not constipation alone. We found limited evidence of associations with socioeconomic background and no evidence of associations with age at initiation of toilet training, breast feeding, gestational age or birth weight.

**Conclusion** Constipation alone was the most prevalent pattern in this cohort. Treatment for hard stools in early childhood is needed to prevent chronic constipation at school age. Constipation with soiling was less common than soiling alone. Further research is needed into the causes of non-retentive soiling.

## What is already known on this topic?

► Constipation and soiling are common in childhood, but little is known about factors that increase the risk of these problems persisting at primary school age.
► The majority of earlier studies examining risk factors for childhood constipation and soiling are cross-sectional, and many are based on small/clinic samples.
► No prospective cohort studies have examined the association between risk factors in early childhood and different trajectories of constipation and soiling at primary school age.

## What this study hopes to add?

► This study finds evidence that risk factors in early childhood are differentially associated with different trajectories of childhood constipation and soiling.
► Contrary to common beliefs of clinicians, we found that among children with soiling those with soiling alone (ie, without constipation) outnumbered those with associated constipation.
► Experiencing hard stools in early childhood is a risk factor for constipation; developmental delay is a risk factor for soiling alone and constipation with soiling.

## INTRODUCTION

Bowel problems are common in childhood and have a considerable impact on quality of life.[1] It is believed that 80% of faecal incontinence is due to overflow from chronic constipation, while 20% have no constipation (functional non-retentive faecal incontinence).[2] The Rome-IV definition for functional constipation at developmental age ≥4 years requires at least two of six symptoms (two or fewer defecations in the toilet per week; at least one episode of faecal incontinence per week; history of retentive posturing/stool retention; history of painful or hard stools; a large faecal mass in the rectum; large diameter stools that can obstruct the toilet) present once a week or more for at least 1 month.[3] Rome-IV diagnostic criteria are also available for functional constipation in children under 4 years.[4] The diagnostic criteria for functional non-retentive faecal incontinence are inappropriate defecation; no medical condition for symptoms and no retention (criteria should be met for at least 1 month).[3] Other clinical definitions are sometimes used.[5 6] Epidemiological studies of the prevalence of constipation and soiling vary probably because of different

definitions. A systematic review reported the median prevalence of constipation in children aged 0–18 years to be 8.9%, with similar prevalence in boys and girls;[5] however, more recent findings suggest a higher proportion of constipation in girls.[6] The prevalence of childhood soiling is between 1% and 4% and is consistently found to be two to four times more common in boys.[7 8] A recent large cross-sectional study of children aged 5–13 years reported that 7.8% (9.8% boys, 5.8% girls) experienced faecal incontinence.[9] Only one epidemiological survey, of children aged 10–16 years in Sri Lanka, differentiated between soiling with and without constipation and reported that 2.0% experienced faecal incontinence and 18% of those did not have constipation.[10]

Early identification of children at risk of constipation and soiling could lead to timely interventions to reduce the adverse impacts on quality of life and psychosocial development. Clinicians believe that pain of passing hard stools in infancy and early childhood is the principal contributing factor for acute childhood constipation,[11] leading to chronic constipation which causes soiling.[11 12] Hard stools lead to withholding and toileting refusal,[13] retaining a stool mass and increasing the difficulty of evacuating. Breastfed infants produce softer stools,[14] and those breastfed for <6 months may develop constipation more commonly.[15] Other risk factors include lower levels of parental education,[9 16] income[9] and socioeconomic status,[8 10] low birth weight and prematurity[17] and developmental delay.[18] Timing of toilet training has also been investigated but findings are inconsistent.[7 19 20] Only one study specified whether constipation occurred with or without soiling.[11] Very little is known about risk factors for soiling without constipation. Finally, most earlier studies of risk factors for constipation and soiling are cross-sectional which makes the timing of events more difficult to determine.

Although most children achieve bowel control by 3–5 years,[19 21] there is recent evidence for different patterns of development of bowel control.[22] These 'developmental trajectories' distinguish children with normative development (89.0%), delayed attainment (4.1%), persistent soiling (2.7%) and relapses in soiling (4.1%).[22] Describing developmental trajectories of soiling alone does not allow the determination of whether soiling is occurring with or without constipation. The aims of this paper are twofold: first, we extend previous work using data from the Avon Longitudinal Study of Parents and Children (ALSPAC) birth cohort to determine the degree of comorbidity between constipation and soiling in childhood, and second, we examine the association between risk factors in early childhood and trajectories of constipation and soiling at primary school age.

## METHODS
### Participants
The sample comprised participants from the ALSPAC. Detailed information is available at http://www.bristol.ac.uk/alspac, including a fully searchable data dictionary http://www.bris.ac.uk/alspac/researchers/data-access/data-dictionary. Pregnant women resident in the former Avon Health Authority in southwest England, having an estimated date of delivery between 1 April 1991 and 31 December 1992 were invited to take part, resulting in a cohort of 14 541 pregnancies.[23] Of the 13 978 singletons/twins alive at 1 year, 24 participants withdrew consent, leaving a starting sample of 13 954.

### Soiling and constipation in mid-childhood
When children were 4½, 5½, 6½, 7½ and 9½ years, their parents were asked, 'How often usually does your child dirty his/her pants during the day?' with options: 'never', 'occasional accident but less than once/week', 'about once/week', '2–5 times/week', 'nearly every day' and 'more than once a day'. Responses were collapsed ('never' vs all other responses) to indicate presence of soiling at each age. Parents were also asked about their child's constipation across a similar age range: 'Has he/she had any constipation in the past 12 months?' with options: 'Yes, and saw a doctor'; 'Yes, but did not see the doctor' and 'No, did not have'. Responses were collapsed (yes with/without consultation vs no) to indicate presence of constipation at each age.

### Risk factors in early childhood
Potential risk factors were identified from the literature. Data were obtained from questionnaires completed by mothers and included the child's stool consistency at 2½ years, breast feeding during the first six months, family socioeconomic position assessed during the antenatal period (parental social class, maternal educational attainment) or at 33 months (material hardship, home ownership and car access), length of gestation and birth weight, developmental level[24] and age at initiation of toilet training (assessed when the child was 6, 15 and 24 months).

### Statistical modelling
We have previously used longitudinal Latent Class Analysis (LLCA) to derive developmental trajectories of soiling[22] where we showed that patterns of soiling at 4–9 years could be adequately explained by a four-class solution. However, such a model ignores the comorbidity with constipation. Therefore, in the current study we (i) estimated a similar latent class model of constipation and (ii) combined this with soiling in a parallel model to describe changes in both problems (using Mplus V.7.11[25]). Further details regarding the estimation can be found in the online supplementary appendix. In addition, as various sample sizes have been used for the different analytical steps online supplementary figure 1 shows a flow chart describing how each sample was obtained.

Once the optimal model had been established, we examined the association between the risks (above) and constipation/soiling class membership using multinomial

**Table 1** Prevalence of soiling and constipation in the Avon Longitudinal Study of Parents and Children study

| | 4½ years | 5½ years | 6½ years | 7½ years | 9½ years | |
|---|---|---|---|---|---|---|
| *Subset of dataset containing at least one non-missing time points for both soiling and constipation (max n=10 450)* | | | | | | |
| No soiling | 8720 (92.8%) | 8284 (93.8%) | 7752 (92.2%) | 7617 (93.1%) | 7288 (94.8%) | |
| Soiling | 673 (7.2%) | 548 (6.2%) | 654 (7.8%) | 565 (6.9%) | 397 (5.2%) | |
| | 4 years 9 months | 5 years 9 months | 6 years 9months | 7 years 7 months | 8 years 7 months | 10 years 8 months |
| No constipation | 7947 (85.6%) | 7622 (89.6%) | 7580 (89.7%) | 7346 (89.9%) | 7062 (90.2%) | 6631 (90.4%) |
| Constipation | 1342 (14.4%) | 881 (10.4%) | 873 (10.3%) | 822 (10.1%) | 765 (9.8%) | 707 (9.6%) |
| *Subset of dataset containing at least three non-missing time points for both soiling and constipation (max n=8435)* | | | | | | |
| No soiling | 7507 (92.8%) | 7518 (94.0%) | 7268 (92.2%) | 7153 (93.2%) | 6801 (94.8%) | |
| Soiling | 585 (7.2%) | 477 (6.0%) | 611 (7.8%) | 525 (6.8%) | 372 (5.2%) | |
| | 4 years 9 months | 5 years 9 months | 6 years 9 months | 7 years 7 months | 8 years 7 months | 10 years 8 months |
| No constipation | 6893 (85.5%) | 7014 (89.7%) | 7086 (89.6%) | 6894 (90.0%) | 6549 (90.5%) | 6194 (90.6%) |
| Constipation | 1173 (14.5%) | 809 (10.3%) | 825 (10.4%) | 766 (10.0%) | 689 (9.5%) | 644 (9.4%) |
| *Sample with complete data on all eleven measures (n=4931)* | | | | | | |
| No soiling | 4575 (92.8%) | 4638 (94.1%) | 4568 (92.6%) | 4590 (93.1%) | 4670 (94.7%) | |
| Soiling | 356 (7.2%) | 293 (5.9%) | 363 (7.4%) | 341 (6.9%) | 261 (5.3%) | |
| | 4 years 9 months | 5 years 9 months | 6 years 9 months | 7 years 7 months | 8 years 7 months | 10 years 8 months |
| No constipation | 4244 (86.1%) | 4425 (89.7%) | 4402 (89.3%) | 4426 (89.8%) | 4454 (90.3%) | 4458 (90.4%) |
| Constipation | 687 (13.9%) | 506 (10.3%) | 529 (10.7%) | 505 (10.2%) | 477 (9.7%) | 473 (9.6%) |

logistic regression. Coefficients and SEs were estimated using the bias-adjusted three-step approach[26 27] which has been shown to reduce the bias inherent in such regression models.[26] For most risks, we report univariable estimates of association; however for toilet training initiation, we considered the potential confounding effects of gender, social class, early parenthood, mother's education, housing adequacy, major financial difficulties, family size, social network/support, developmental delay and maternal depression.

## RESULTS
Table 1 shows rates of soiling and constipation at each age. While rates of reported constipation decrease steadily, there is a more complex pattern for soiling. Of those reporting constipation, the proportion who saw a doctor decreased from 19.7% to 11.6% (data not shown).

### Unconditional model for constipation
There was good support for a four-class solution when considering the six repeated measurements of constipation. Focusing on the sample consisting of cases with at least three non-missing measures of constipation (n=8979), the following classification of children was obtained: normative (82%; very low probability of constipation throughout), early childhood occurrence (7%; children suffering from constipation until 6 years), late childhood occurrence (8% problem with constipating emerging after 6 years) and persistent (3%; high probability of problems throughout).

### Parallel model for soiling and constipation
The next step was to merge the two LLCA models into a single parallel model permitting an investigation into the longitudinal association between soiling and constipation (see table 2 and figure 1). Based on these results, we collapsed the 16 groups defined by such a two-way classification into four clinical relevant subgroups to facilitate further study. The four resulting groups were: *normative* (74.5% of total)—normative classes for both constipation and soiling; *constipation alone* (13.2%)—normative class for soiling and non-normative classes for constipation; *soiling alone* (7.5%)—normative class for constipation and non-normative classes for soiling; *constipation with soiling* (4.8%)—classified as non-normative in terms of the progression of both constipation and soiling during the period of childhood studied.

### Rates of associated symptoms within each class
To gain further insight into this classification, we examined the extent to which the rates of daytime wetting, bed-wetting and stomach ache in mid-childhood, and infrequent bowel movements in early childhood differed across the four composite classes (online supplementary figure 2). First, the two classes involving soiling exhibited similar rates of both daytime wetting and bed-wetting, and furthermore, these rates were consistently higher than the constipation only and the normative class. Second, rates of stomach ache were lower for the soiling alone class compared with either class reporting constipation. Finally, in terms of the rates of infrequent bowel movements in early childhood, those classified as soiling alone were no different to the normative class for

**Table 2** Associations between constipation and soiling: joint distribution of classes derived from parallel longitudinal Latent Class Analysis model (n=8435)

| Constipation classes | Soiling classes (%) | | | | |
|---|---|---|---|---|---|
| | Normative (soiling) | Delayed | Relapse | Persistent | Total |
| Normative (constipation) | 74.5* | 4.3† | 2.2† | 1.0† | 82 |
| Early childhood occurrence | 5.5‡ | 1.1§ | <0.1§ | 0.5§ | 7.1 |
| Late childhood occurrence | 5.8‡ | 0.6§ | 0.9§ | 0.3§ | 7.6 |
| Persistent | 1.9‡ | 0.4§ | 0.5§ | 0.5§ | 3.3 |
| Total | 87.7 | 6.4 | 3.6 | 2.3 | 100 |

Data shown in the table are class proportions for the latent class patterns based on the estimated model. These results are for the sample of children with both constipation and soiling data for at least three time points.
*Normative (74.5% of total)—normative classes for both constipation and soiling.
†Soiling alone (7.5% of total)—normative class for constipation and non-normative classes for soiling.
‡Constipation alone (13.2% of total)—normative class for soiling and non-normative classes for constipation.
§Constipation with soiling (4.8% of total)—non-normative classes for constipation and soiling.

whom infrequent movements were rare, whereas both classes experiencing constipation were seen to have infrequent bowel movements which became a more common finding as the children became older.

### Risk factors for constipation/soiling class membership

Table 3 shows the association between the risk factors and ORs for membership of the classes of 'constipation alone', 'soiling alone' and 'constipation with soiling' with reference to the normative class. For completeness, we append the results for the separate models of soiling and constipation (online supplementary table 3).

There was strong evidence that boys were at greater odds of being a member of the 'soiling alone' and 'constipation with soiling' classes; however, at lower odds of experiencing constipation alone. We also found strong evidence that the presence of hard stools (sometimes or usually) at 2½ years was associated with increased odds of membership to the 'constipation alone' class; however,

there was little evidence of hard stool consistency being associated with constipation with soiling. There was also a suggestion that hard stools might be protective against the development of 'soiling without constipation'.

We found little evidence of an association between breastfeeding duration and later problems with constipation and/or soiling, and we found no association between gestational age or birth weight and constipation and soiling classes. For socioeconomic position, the pattern of results was inconsistent. We found no evidence that parental social class, maternal educational attainment and home ownership were associated with constipation and soiling. Material hardship, however, was associated with constipation with soiling and lack of car access was associated with soiling alone. Lower developmental level at 18 months was associated with 'soiling alone' and 'constipation and soiling'. However, there was no association between developmental level 'constipation alone'.

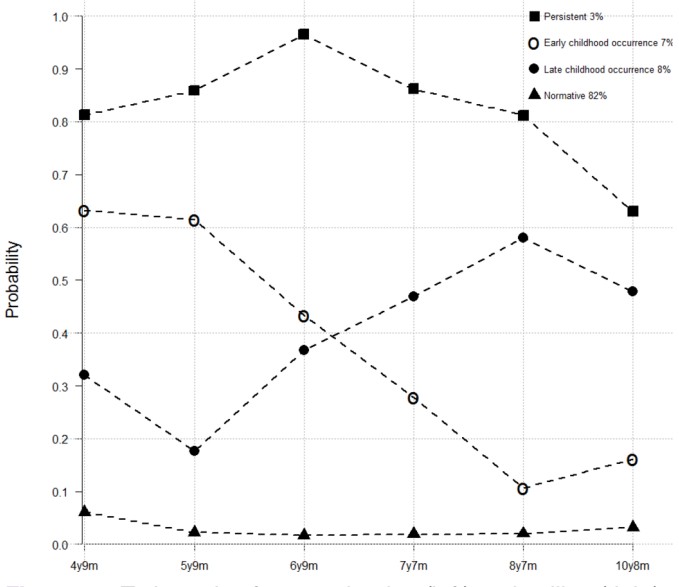
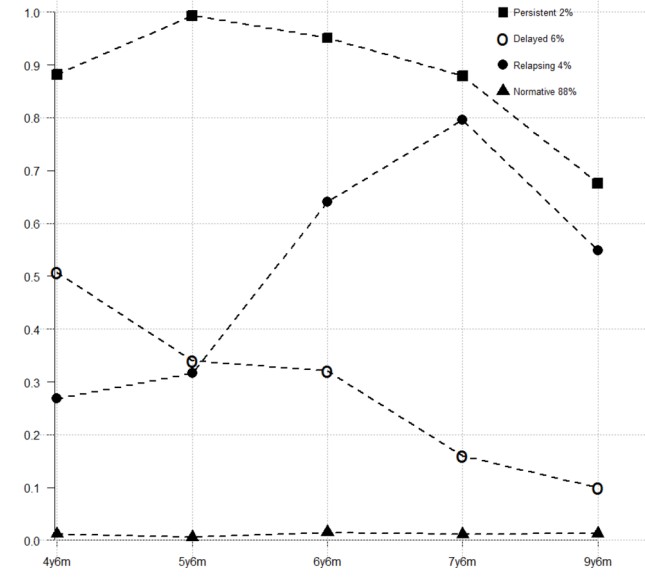

**Figure 1** Trajectories for constipation (left) and soiling (right).

**Table 3** Association between risk factors and the parallel classes of soiling and constipation (n≤8435 depending on risk factor)

| | n (%) | Constipation alone OR (95% CI) | Soiling alone OR (95% CI) | Constipation with soiling OR (95% CI) | Omnibus P value |
|---|---|---|---|---|---|
| **Sex** | | | | | |
| Female (ref) | 4082 (52%) | | | | <0.001 |
| Male | 4353 (48%) | 0.63 (0.53 to 0.75) | 1.78 (1.39 to 2.27) | 1.38 (1.06 to 1.79) | |
| **Hard stools regularity at 2½ years** | | | | | |
| Never hard (ref) | 1414 (18%) | | | | <0.001 |
| Sometimes | 4312 (54%) | 2.08 (1.56 to 2.77) | 1.12 (0.82 to 1.52) | 1.22 (0.84 to 1.78) | |
| Usually hard | 2235 (28%) | 1.90 (1.40 to 2.60) | 0.68 (0.46 to 0.99) | 1.26 (0.85 to 1.89) | |
| **Breast feeding** | | | | | |
| Child never breast fed (ref) | 1695 (21%) | | | | 0.309 |
| Child breast fed for <6 months | 3579 (45%) | 1.25 (0.99 to 1.58) | 1.34 (0.96 to 1.86) | 1.08 (0.77 to 1.52) | |
| Child breastfed for at least 6 months | 2677 (34%) | 1.24 (0.97 to 1.59) | 1.22 (0.86 to 1.73) | 0.94 (0.65 to 1.35) | |
| **Parental social class** | | | | | |
| Professional, managerial or skilled | 6579 (85%) | | | | 0.736 |
| Partly skilled/unskilled | 1178 (15%) | 0.95 (0.74 to 1.20) | 1.16 (0.85 to 1.60) | 0.98 (0.67 to 1.42) | |
| **Maternal educational attainment** | | | | | |
| A level/degree (ref) | 3399 (41%) | | | | 0.358 |
| O level | 2916 (35%) | 0.92 (0.76 to 1.12) | 0.82 (0.62 to 1.08) | 1.16 (0.87 to 1.54) | |
| Vocational or none | 1909 (23%) | 0.82 (0.66 to 1.03) | 0.94 (0.70 to 1.26) | 0.92 (0.65 to 1.30) | |
| **Material hardship at 33 months*** | | | | | |
| No material hardship (score<5) (ref.) | 5733 (73%) | | | | 0.026 |
| Material hardship (score≥5) | 2084 (27%) | 1.00 (0.82 to 1.22) | 1.06 (0.81 to 1.40) | 1.52 (1.15 to 2.00) | |
| **Home ownership at 33 months** | | | | | |
| Home owned/mortgaged (ref.) | 6524 (85%) | | | | 0.401 |
| Privately rented | 312 (4.1%) | 0.73 (0.44 to 1.20) | 1.12 (0.62 to 2.02) | 1.26 (0.69 to 2.29) | |
| Subsidised rented | 832 (11%) | 0.74 (0.55 to 1.00) | 1.01 (0.68 to 1.48) | 0.92 (0.60 to 1.43) | |
| **Car access at 33 months** | | | | | |
| Yes (ref.) | 7230 (93%) | | | | 0.018 |
| No | 581 (7%) | 0.82 (0.57 to 1.18) | 1.53 (1.02 to 2.27) | 1.49 (0.97 to 2.29) | |
| **Gestational age at delivery** | | | | | |
| ≥37 weeks (ref) | 8000 (95%) | | | | 0.334 |
| <37 weeks | 435 (5%) | 0.76 (0.50 to 1.16) | 1.28 (0.80 to 2.05) | 1.12 (0.66 to 1.93) | |
| **Birth weight** | | | | | |
| ≥2500 g (ref) | 7964 (96%) | | | | 0.488 |
| <2500 g | 372 (4%) | 1.29 (0.88 to 1.88) | 1.32 (0.79 to 2.21) | 1.05 (0.56 to 1.97) | |
| **Developmental level at 18 m†** | | | | | |
| Per 1 SD reduction in development | 7931 | 0.95 (0.87 to 1.05) | 1.44 (1.28 to 1.62) | 1.31 (1.13 to 1.51) | <0.001 |

| | n (%) | Constipation alone OR (95% CI) | Soiling alone OR (95% CI) | Constipation with soiling OR (95% CI) | Omnibus P value |
|---|---|---|---|---|---|
| **Table 3** Continued | | | | | |
| Age at initiation of toilet training | | | | | |
| Unadjusted model | | | | | <0.001 |
| Before 6 months | 168 (2%) | 1.20 (0.67 to 2.15) | 1.50 (0.69 to 3.28) | 1.73 (0.78 to 3.81) | |
| Between 6 and 15 months | 1056 (14%) | 1.22 (0.95 to 1.56) | 1.09 (0.74 to 1.59) | 0.81 (0.49 to 1.32) | |
| Between 15 and 24 months (ref) | 3841 (50%) | | | | |
| After 24 months | 2650 (34%) | 0.91 (0.75 to 1.12) | 1.47 (1.13 to 1.91) | 1.56 (1.17 to 2.07) | |

*Material hardship was assessed using the set of questions: 'How difficult at the moment do you find it to afford these items? Food, clothing, heating, rent, items for child': each on a four-point scale from very difficult through to not difficult. Responses were summed and a binary variable was derived to indicate the top 20% of the sample.
†Developmental level was assessed using a questionnaire developed by Avon Longitudinal Study of Parents and Children including items from the Denver Developmental Screening Test[24] and comprising four domains of development (fine motor, gross motor, communication and social skills). We used a total development score derived from the sum of the scores on each domain. Scores on each domain were adjusted for age in weeks and standardised (using a linear regression model and extracting the residuals) and reversed where appropriate so that high values on all scores reflected a lower level of development (increase in the odds of membership to the latent classes per 1 SD increase in developmental level score).

Finally, a later age at initiation of toilet training was associated with 'soiling alone' and 'constipation with soiling'. Estimates were attenuated following adjustment for the confounders: 'soiling alone' (adjusted OR 1.30 (0.97 to 1.75)), 'constipation with soiling' (adjusted OR 1.27 (0.92 to 1.75)). The most influential confounders were developmental delay, major financial difficulties, sex (male) (membership of 'soiling alone' class) and maternal depression ('constipation with soiling' class).

## DISCUSSION

We found that the variability in longitudinal data on childhood constipation and soiling years in a large UK birth cohort could each be adequately explained by four latent classes. On cross-classifying these groupings we identified four clinically relevant longitudinal classes. Three quarters of children were members of the 'normative' class with a very low probability of constipation or soiling across childhood. Two classes comprised children with constipation alone (13.2% of children) and soiling alone (7.5%), and finally, a class of children who had constipation with soiling (4.8%). Among children with soiling, around 60% had soiling alone and around 40% had soiling with constipation. The prevalence of constipation (18%) observed is higher than the median prevalence (8.9%) reported in a systematic review of children aged 0–18 years.[5] Boys were more likely than girls to experience soiling as reported elsewhere.[5 7]

We found that hard stools in early childhood were associated with an increased odds of 'constipation alone' at school age, as well as a reduced risk for 'soiling alone'.

Children with developmental delay had more 'soiling alone' and 'constipation with soiling', but not 'constipation alone'. We found limited evidence for socioeconomic disadvantage and no evidence that a shorter duration of breast feeding, shorter gestation, lower birth weight or the timing of toilet training were risk factors for constipation or soiling at school age.

A major strength is the availability of repeated measures of constipation and soiling across childhood in a large, representative cohort. Using these data, we extended our previous work by modelling constipation and soiling in parallel. The resultant latent classes allowed us to estimate, the prevalence of soiling with or without constipation and for constipation alone across childhood and the differences in early risk factors.

A potential limitation is the use of maternal report measures of constipation and soiling. Parents were asked to report whether their child had suffered from constipation in the past 12 months and whether they saw a doctor. Previous studies report that only a small proportion of children see a doctor for soiling problems,[8] perhaps because parents are unaware that medical advice and treatment is available. Also, parents of children who soil may be unaware that their child is constipated. In addition, the questions relating to soiling did not elicit information about quantity so did not distinguish between leakage, normal bowel movement and staining. Some parents added addenda to the questionnaire noting, for instance, active diarrhoea and poor wiping and did not categorise their child as soiling, so anecdotal evidence indicates that parents are able to distinguish between true soiling and other occurrences.

Parents in the present study were not asked about the duration of constipation or soiling at each time point; however, the repeated measurements of these conditions suggest that constipation and soiling are persisting problems. Information on frequency of soiling was omitted from the latent class models because frequent soiling was rare. Our aim was to describe the trajectories of constipation and soiling in the community and not to focus solely on children whose bowel problems meet the current established diagnostic criteria. Finally, there was no information available on underlying organic causes of constipation and soiling (eg, Hirschsprung's disease or anorectal malformations), but the majority of cases are functional. Children with soiling and/or constipation have relatives with these problems,[7 28 29] information we did not have.

Clinicians believe that most cases of childhood soiling are from chronic constipation; however, we found that among soilers, constipation with soiling (39%) was less common than soiling alone (61%). Other estimates for constipation with soiling[2 28] are from clinic samples. Children with soiling present less often to a clinician, perhaps because they feel ashamed or they believe it is due to laziness.[8] Compared with constipation with soiling, which comes with abdominal pain and infrequent stools, soiling may have no other symptoms.[2] In agreement with earlier studies,[11 12] we found that hard stools in early childhood were strongly associated with later constipation. Early hard stools were, however, not associated with soiling alone or constipation *with* soiling, suggesting that soiling is a primary continence issue and not secondary to constipation.

There is evidence from earlier studies that constipation and soiling are more common among children from lower socioeconomic background.[8–10 16] We found little evidence that socioeconomic factors are associated with childhood constipation and soiling. Developmental delay was associated with any soiling but not with constipation alone. Constipation alone may be more strongly related to family history, early experiences of painful defecation and defecation anxiety while soiling is related to delays in reaching social developmental milestones. We found no evidence that length of gestation or birth weight was associated with constipation or soiling at school age consistent with an earlier study.[19] After adjusting for confounders, earlier or later initiation of toilet training had no significant effect on constipation or soiling similar to previous studies[19 20] and in contrast to a large cohort study.[7]

This study finds evidence that experiencing hard stools in early childhood is a risk factor for later problems with constipation at school age. Early identification of children at risk of constipation is important because more than a third of cases become chronic and require secondary care.[30] Early diagnosis and treatment will reduce the risk of constipation persisting to school age, resulting in poor quality of life. The burden of constipation is large—13.2% of primary school-age children. Extra healthcare costs are substantial as 5.9% of the ALSPAC cohort 4–9 years of age

saw a doctor at least once (data not shown). Many would have been referred to secondary care and tertiary paediatric gastroenterology and paediatric surgery services.

Further population-based studies are needed, and if non-retentive soiling rates are confirmed, parents should be encouraged to seek help for soiling from clinicians who recognise the problem. As with daytime wetting the mainstay of treatment for soiling (alone and the soiling aspect of soiling with constipation) will be behavioural, for example, supporting children to attempt to empty their bowels into the toilet at least once per day. This form of therapy requires practitioners to be skilled at providing behavioural interventions concurrently with medications for children who are constipated.

**Acknowledgements** The authors are extremely grateful to all the families who took part in this study, the midwives for their help in recruiting them and the whole ALSPAC team, which includes interviewers, computer and laboratory technicians, clerical workers, research scientists, volunteers, managers, receptionists and nurses.

**Contributors** Secured funding: DT and CJ. Study design: JH, DT, AvG and CJ. Literature search: CJ. Data analysis: JH and MG. Data interpretation and writing: JH, MG, DT, AvG and CJ.

**Funding** The MRC and Wellcome Trust (grant ref: 102215/2/13/2) and University of Bristol provide core support for ALSPAC. This publication is the work of the authors who will serve as guarantors for the contents of this paper. This work was supported by the MRC (MR/L007231/1) and NHS Greater Glasgow & Clyde Health Board (Stephen McLeod, Head of Specialist Children's Services).

**Competing interests** None declared.

**Ethics approval** ALSPAC Law and Ethics committee and local research ethics committees.

**Provenance and peer review** Not commissioned; externally peer reviewed.

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
