## [Reviewer comments · BMJ Paediatrics Open]

ARTICLE DETAILS

TITLE (PROVISIONAL)	Early childhood risk factors for constipation and soiling at school age: an observational cohort study
AUTHORS	Heron, Jon; Grzeda, Mariusz; Tappin, David; von Gontard, Alexander; Joinson, Carol

VERSION 1 - REVIEW

REVIEWER	Nevitt, Sarah University of Liverpool, United Kingdom Competing interests: I have no competing interests
REVIEW RETURNED	06-Nov-2017

GENERAL COMMENTS	I have provided a statistical review of the manuscript “Early childhood risk factors for constipation and soiling at school age: an observational cohort study.” This manuscript presents a latent class analysis based on longitudinal patterns of constipation and soiling within a large cohort study of nearly 9000 participants. The authors are experienced in the statistical methods used within this manuscript, having previously used the methodology to derive a four-class solution for soiling. The methodology is now extended to a joint / parallel model into a two-way classification, each of four classes for both constipation and soiling. The authors present extensive details of their methodology within the supplementary material and all methodology appears to have been used appropriately and results are very well described and discussion well written. I just have a couple of minor suggestions for the authors, based on a couple of areas of uncertainty when reading the manuscript: 1) Methods (page 7, lines 36-37 and lines 44-45): “Responses were collapsed to indicate the presence of soiling / constipation at each age.” Please indicate how this was done. For example, it is not immediately clear to me if “Occasional accident but less than once/week” would be classed as soiling or not. 2) The results are presented very clearly but the only thing I found slightly confusing was all of the different numbers of participants contributing to different analyses and tables – e.g. n=8,979 or n=8,435. I suggest for complete clarity adding another table in the Supplementary material which describes all of these samples and where they are used in the analyses. 3) For toilet training initiation, where an adjusted analysis has been performed, I would say it is appropriate to report numerical results (e.g. the odds ratios). However, for the Univariable analyses, I would report only the direction of the association, rather than
--

	numbers due to likely associations and confounding between the risk factors within Table 3 (e.g. page 15, line 19-20 “a reduction in odds of approximately one third.”) 4) Discussion (page 16, line 15). Is ‘phenotypes’ an appropriate word to use here given the genetic associations with this word when as far as I can tell, genetic considerations have not been made here? Perhaps just state ‘classes’ instead? 5) I may be wrong here as I am not a clinical expert but could it be classed as a ‘limitation’ or ‘unanswered question’ that the data used here is from a cohort study in England and results may not be applicable to other countries / settings? For example, the authors describe a study from Sri Lanka within the introduction and I wonder how comparable the two cohorts of participants would be?
--	---

REVIEWER	Wallace, Chris Pennine Acute NHS Trust UK Competing interests: Nil
REVIEW RETURNED	27-Nov-2017

GENERAL COMMENTS	Well written, clear in methodology and reporting results. Interesting possible implications for practice. I would suggest that caution should be taken before interpreting the finding about a lack of association between hard stool and overflow, as you mentioned in the paper parental reporting may mean some overflow related to unacknowledged preceding constipation may be missed. However, this point is picked up and explored in the 'Strengths and Weaknesses' section, and so is suitably covered. The single minor revision I would suggest is that a small addendum to your 'Strengths and Weaknesses' section should briefly mention that this data was from a single health authority (albeit with a large data set). In order to extrapolate nationally it would be helpful to have data from other regions, as alluded to in the 'Unanswered questions and future research section'.
---

REVIEWER	Baldissera, Marilisa HSL-PUCRS, Brazil Competing interests: nothing to disclose
REVIEW RETURNED	31-Dec-2017

GENERAL COMMENTS	It is a good job and can contribute to daily practice.
--

VERSION 1 – AUTHOR RESPONSE

Reviewer: 1

This manuscript presents a latent class analysis based on longitudinal patterns of constipation and soiling within a large cohort study of nearly 9000 participants.

The authors are experienced in the statistical methods used within this manuscript, having previously used the methodology to derive a four-class solution for soiling. The methodology is now extended to a joint / parallel model into a two-way classification, each of four classes for both constipation and soiling. The authors present extensive details of their methodology within the supplementary material and all methodology appears to have been used appropriately and results are very well described and discussion well written.

I just have a couple of minor suggestions for the authors, based on a couple of areas of uncertainty when reading the manuscript:

1) Methods (page 7, lines 36-37 and lines 44-45): “Responses were collapsed to indicate the presence of soiling / constipation at each age.”

Please indicate how this was done. For example, it is not immediately clear to me if “Occasional accident but less than once/week” would be classed as soiling or not.

This has been clarified in the text

2) The results are presented very clearly but the only thing I found slightly confusing was all of the different numbers of participants contributing to different analyses and tables – e.g. n=8,979 or n=8,435. I suggest for complete clarity adding another table in the Supplementary material which describes all of these samples and where they are used in the analyses.

We agree that the different sample sizes are a little confusing. This is caused by our perceived need to derive the mixture-model for constipation initially using all available data thus rendering the output from that first step usable in a broader range of future work. In the hope of clarifying the situation we have added a flow-chart (Figure S1) to the supplementary materials.

3) For toilet training initiation, where an adjusted analysis has been performed, I would say it is appropriate to report numerical results (e.g. the odds ratios). However, for the Univariable analyses, I would report only the direction of the association, rather than numbers due to likely associations and confounding between the risk factors within Table 3 (e.g. page 15, line 19-20 “a reduction in odds of approximately one third.”)

We have removed all mention of the magnitude of univariable parameter estimates.

4) Discussion (page 16, line 15). Is 'phenotypes' an appropriate word to use here given the genetic associations with this word when as far as I can tell, genetic considerations have not been made here? Perhaps just state 'classes' instead?

We agree with this point and have changed the text accordingly.

5) I may be wrong here as I am not a clinical expert but could it be classed as a 'limitation' or 'unanswered question' that the data used here is from a cohort study in England and results may not be applicable to other countries / settings? For example, the authors describe a study from Sri Lanka within the introduction and I wonder how comparable the two cohorts of participants would be?

Point also raised by reviewer 2 – response given below

Reviewer: 2

Well written, clear in methodology and reporting results. Interesting possible implications for practice.

I would suggest that caution should be taken before interpreting the finding about a lack of association between hard stool and overflow, as you mentioned in the paper parental reporting may mean some overflow related to unacknowledged preceding constipation may be missed. However, this point is picked up and explored in the 'Strengths and Weaknesses' section, and so is suitably covered.

We agree with this cautionary note and thank the reviewer for the comment.

The single minor revision I would suggest is that a small addendum to your 'Strengths and Weaknesses' section should briefly mention that this data was from a single health authority (albeit with a large data set). In order to extrapolate nationally it would be helpful to have data from other regions, as alluded to in the 'Unanswered questions and future research section'.

The literature indicates that constipation is a global phenomenon. Of course, studies from other regions would add weight, but the unique nature of this dataset is that it follows individuals over time where data was collected not specifically related to a 'constipation' service and therefore covers all

children born in the area. Also, the data is collected during the periods of change from breast to formula milk, from purely milk to the addition of solids and from incontinence to continence (the poorly named period of 'toilet training').

We note that in the final paragraph we do state the need for further studies.

Reviewer: 3

It is a good job and can contribute to daily practice.

We thank the reviewer for this positive response